# WHAT DOES STABLE DIFFUSION KNOW ABOUT THE 3D SCENE?

## ABSTRACT

Recent advances in generative models like Stable Diffusion (Rombach et al., 2022) enable the generation of highly photo-realistic images. Our objective in this paper is to probe the diffusion network to determine to what extent it 'understands' different properties of the 3D scene depicted in an image. To this end, we make the following contributions: (i) We introduce a protocol to evaluate whether a network models a number of physical 'properties' of the 3D scene by probing for explicit features that represent these properties. The probes are applied on datasets of real images with annotations for the property. (ii) We apply this protocol to properties covering scene geometry, scene material, support relations, lighting, and view dependent measures. (iii) We find that Stable Diffusion is good at a number of properties including scene geometry, support relations, shadows and depth, but less performant for occlusion. (iv) We also apply the probes to other models trained at large-scale, including DINO, CLIP and VQGAN, and find their performance inferior to that of Stable Diffusion.

## 1 INTRODUCTION

Image generation with diffusion models (Sohl-Dickstein et al., 2015), following on from earlier generation using GANs (Goodfellow et al., 2014), has achieved amazing results in terms of verisimilitude (Rombach et al., 2022). This naturally raises the question: *to what extent does the diffusion network 'understand' (or model) the 3D scene depicted in the image?* For example, does the network *implicitly* have an image rendering pathway that models 3D geometry and surfaces, and then projects to generate an image taking account of occlusion and perspective? As an indication that the diffusion network is 3D and physics aware, Figure 1 shows the result of the off-the-shelf Stable Diffusion model (Rombach et al., 2022) inpainting masked regions in real images – it correctly predicts shadows and supporting structures.

To answer this question, we propose an evaluation protocol to systematically 'probe' a diffusion network on its ability to represent a number of 'properties' of the 3D scene and viewpoint. These properties include: 3D structure and material of the scene, such as surface layout; lighting, such as object-shadow relationships; and viewpoint dependent relations such as occlusion and depth.

The protocol involves three steps: *First*, a suitable image evaluation dataset is selected that contains ground truth annotations for the property of interest, for example the SOBA dataset (Wang et al., 2020) is used to probe the understanding of lighting, as it has annotations for object-shadow associations. This provides a train/val/test set for that property; *Second*, a grid search is carried out over the layers and time steps of the diffusion model to select the optimal feature for determining that property. The selection involves learning the weights of a simple linear classifier for that property (e.g. 'are these two regions in an object-shadow relationship or not'); *Third*, the selected feature (layer, time step) and trained classifier are evaluated on a test set, and its performance answers the question of how well the diffusion model 'understands' that property. This protocol could also be equally applied to other pre-trained models.

Specifically, we train and evaluate on real images, inspired by (Tang et al., 2023), we add noise to the input image in the latent space, and compute features from different layers and time steps with an off-the-shelf Stable Diffusion model. While probing the properties, linear classifiers are used to infer the relationships between *regions*, rather than points. The region representation is obtained by a simple average pooling of the diffusion features over the annotated region or object.

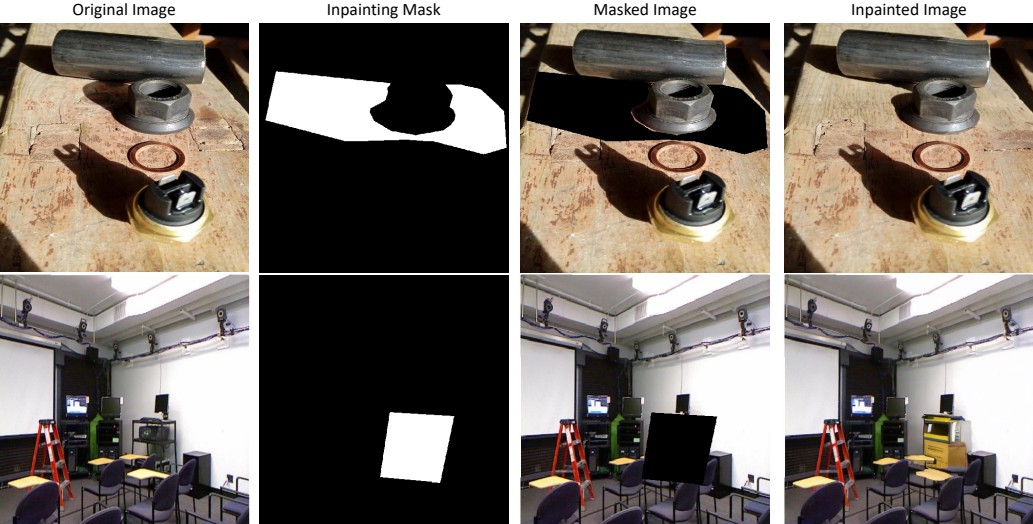

Figure 1: **Motivation: What does Stable Diffusion know about the 3D scene?** The Stable Diffusion inpainting model is here tasked with inpainting the masked region of the real images. It correctly predicts a shadow consistent with the lighting direction (top), and a supporting structure consistent with the scene geometry (bottom). This indicates that the Stable Diffusion model generation is consistent with the geometry (of the light source direction) and physical (support) properties. **Note:** these examples are only for illustration and inpainting is *not* the objective of this paper. Rather, we probe a general Stable Diffusion model to determine whether there are explicit features for such 3D scene properties. The supplementary provides more examples of Stable Diffusion's capability to predict different physical properties of the scene.

From our investigation, we make two observations: *First*, the Stable Diffusion model has a good understanding of the scene geometry, support relations, the lighting, and the depth of a scene. However, material and occlusion understanding is challenging for it; *Second*, Stable Diffusion generally demonstrates better performance for 3D properties, than other strong self-supervised features, such as OpenCLIP, DINOv1 and DINOv2, and other strong generative models, such as VQGAN. The DINO model has been well known for being good at semantic image segmentation (Melas-Kyriazi et al., 2022; Shin et al., 2022; Siméoni et al., 2021). Our findings open up the possibility of using features from Stable Diffusion in downstream tasks where they are stronger than those of DINO.

We describe the properties explored, the protocol, datasets and classifiers in Section 3. Experimental results of both Stable Diffusion and other pre-trained discriminative and generative models are given in Section 4, and finally future work is discussed in Section 5.

## 2 RELATED WORK

### 2.1 GENERATIVE MODELS

Generative models have made significant achievements in advancing image quality and diversity in the recent literature. A series of generative models, such as Variational Autoencoders (VAEs) (Kingma & Welling, 2014), Generative Adversarial Networks (GANs) (Goodfellow et al., 2014), Flow-based Generators (Dinh et al., 2014), and Diffusion Probabilistic Models (DPMs) (Sohl-Dickstein et al., 2015), have been proposed. These models have contributed to widespread tasks, including image completion (Pathak et al., 2016), composition (Lin et al., 2018), interpolation (Karras et al., 2019) and editing (Chai et al., 2021), image-to-image translation (Isola et al., 2017), multi-modalities translation (Hu et al., 2023), and numerous others. We build upon the diffusion models (Rombach et al., 2022), which have shown state-of-the-art generation quality.

## 2.2 EXPLORATION OF PRE-TRAINED MODELS

Building on the success of deep networks, there has been significant interest from the community to understand what has been learnt by these complex models. On discriminative models, for example, (Zeiler & Fergus, 2014; Mahendran & Vedaldi, 2015) propose inverse reconstruction to directly visualize the acquired semantic information in various layers of a trained classification network; (Zhou et al., 2016; Fong & Vedaldi, 2017; Fong et al., 2019) demonstrate that scene classification networks have remarkable localization ability despite being trained on only image-level labels; and (Erhan et al., 2009; Simonyan et al., 2014; Selvaraju et al., 2017) use the gradients of any target concept, flowing into the final convolutional layer to produce a saliency map highlighting important regions in the image for predicting the concept. In the more recent literature, (Chefer et al., 2021) explores what has been learned in the powerful transformer model by visualizing the attention map. On generative models, researchers have mainly investigated what has been learned in GANs, for example, GAN dissection (Bau et al., 2019) presents an analytic framework to visualize and understand GANs at the unit-, object-, and scene-level; (Wu et al., 2021) analyses the latent style space of StyleGANs (Karras et al., 2019).

## 2.3 EXPLOITATION OF GENERATIVE MODELS

Apart from understanding the representation in pre-trained models, there has been a recent trend for exploiting the learnt feature from generative models, to tackle a series of downstream discriminative tasks. For example, leveraging generative models for data augmentation in recognition tasks (Jahanian et al., 2022; He et al., 2023), semantic segmentation via generative models (Baranchuk et al., 2022; Li et al., 2021; Xu et al., 2023), open-vocabulary segmentation with diffusion models (Li et al., 2023), depth maps estimation based on RGB images (Shi et al., 2022; Noguchi & Harada, 2020). More recently, (Bhattad et al., 2023) search for intrinsic offsets in a pre-trained StyleGAN for a range of downstream tasks, predicting normal maps, depth maps, segmentations, albedo maps, and shading. In contrast to this work, we adopt annotations from different datasets for supervision, rather than employing pre-trained prediction models for supervision. A closely related effort to ours is DIffusion FeaTures (DIFT) (Tang et al., 2023), but it only focuses on computing *correspondences* at the geometric or semantic level between images.

## 2.4 PHYSICAL SCENE UNDERSTANDING

There have been works studying different physical properties for scene understanding, including shadows (Wang et al., 2020; 2021), material (Upchurch & Niu, 2022; Sharma et al., 2023), occlusion (Zhan et al., 2022), scene geometry (Liu et al., 2019), support relations (Silberman et al., 2012) and depth (Silberman et al., 2012). However, these works focus on one or two physical properties, and most of them require training a model for the property in a supervised manner. In contrast, we use a single model to predict multiple properties, and do not train the features.

## 3 METHOD – PROPERTIES, DATASETS, AND CLASSIFIERS

Our goal is to examine the ability of a diffusion model to understand different physical properties of the 3D scene, including: scene geometry, material, support relations, shadows, occlusion and depth. Specifically, we conduct linear probing of the features from different layers and time steps of the Stable Diffusion model. First, we set up the questions for each property (Section 3.1); and then select real image datasets with ground truth annotations for each property (Section 3.2). We describe how a classifier is trained to answer the questions, and the grid search for the optimal time step and layer to extract a feature for predicting the property in Section 3.3.

### 3.1 PROPERTIES AND QUESTIONS

Here, we study the diffusion model's ability to predict different *properties* of the 3D scene; the properties cover the 3D structure and material, the lighting, and the viewpoint. For each property, we propose *questions* that classify the relationship between a pair of *Regions*, *A* and *B*, in the same image, based on the features extracted from the diffusion model. The properties and questions are:

Table 1: **Overview of the datasets and training/evaluation statistics for the properties investigated.** For each property, we list the image dataset used, and the number of images for the train, val, and test set. 1000 images are used for testing if the original test set is larger than 1000 images. Regions are selected in each image, and pairs of regions are used for all the probe questions.

| Property: | | Same Plane | Perpendicular Plane | Material | Support Relation | Shadow | Occlusion | Depth |
|---|---|---|---|---|---|---|---|---|
| Dataset: | | ScanNetv2 | ScanNetv2 | DMS | NYUv2 | SOBA | Sep. COCO | NYUv2 |
| Images | # Train | 50 | 50 | 50 | 50 | 50 | 50 | 50 |
| | # Val | 20 | 20 | 20 | 20 | 20 | 20 | 20 |
| | # Test | 1000 | 1000 | 1000 | 654 | 160 | 820 | 654 |
| Regions | # Train | 855 | 489 | 641 | 1040 | 634 | 641 | 1074 |
| | # Val | 390 | 223 | 238 | 440 | 180 | 247 | 457 |
| | # Test | 14913 | 8310 | 11364 | 14008 | 1176 | 4011 | 14707 |
| Pairs | # Train | 2516 | 3104 | 2268 | 1616 | 1268 | 2220 | 3060 |
| | # Val | 1172 | 1396 | 920 | 688 | 360 | 636 | 1282 |
| | # Test | 45076 | 50216 | 41824 | 21768 | 2352 | 6292 | 42026 |

1. *Same Plane*: 'Are Region *A* and Region *B* on the same plane?'

2. *Perpendicular Plane*: 'Are Region *A* and Region *B* on perpendicular planes?'

3. *Material*: 'Are Region *A* and Region *B* made of the same material?'

4. *Support Relation*: 'Is Region *A* (object *A*) supported by Region *B* (object *B*)?'

5. *Shadow*: 'Are Region *A* and Region *B* in an object-shadow relationship?'

6. *Occlusion*: 'Are Region *A* and Region *B* part of the same object but, separated by occlusion?'

7. *Depth*: 'Does Region *A* have a greater average depth than Region *B*?'

We choose these properties as they exemplify important aspects of the 3D physical scene: the *Same Plane* and *Perpendicular Plane* questions probe the 3D scene geometry; the *Material* question probes what the surface is made of, *e.g.,* metal, wood, glass, or fabric, rather than its shape; the *Support Relation* probes the physics of the forces in the 3D scene; the *Shadow* question probes the lighting of the scene; the *Occlusion* and *Depth* questions depend on the viewpoint, and probe the disentanglement of the 3D scene from its viewpoint.

## 3.2 DATASETS

To study the different properties, we adopt various off-the-shelf real image datasets with annotations for the different properties, where the dataset used depends on the property. We repurpose each dataset to support probe questions of the form: $\mathcal{D} = \{(R_A, R_B, y)_1, \ldots, (R_A, R_B, y)_n\}$, where $R_A$, $R_B$ denote a pair of regions, and $y$ is the binary label indicating the answer to the considered question of the probed property. For each property, we create a train/val/test split from those of the original datasets, if all three splits are available. While for dataset with only train/test splits available, we divide the original train split into our train/val splits. Table 1 summarises the datasets used and the statistics of the splits used. We discuss each property and dataset in more detail next.

**Same Plane.** We use the ScanNetv2 dataset (Dai et al., 2017) with annotations for planes from (Liu et al., 2019). Regions are obtained via splitting plane masks into several regions. A pair of regions are *positive* if they are on the same plane, and *negative* if they are on different planes. First row of Figure 2 is an example.

**Perpendicular Plane.** We use the ScanNetv2 dataset (Dai et al., 2017). We use the annotations from (Liu et al., 2019) which provide segmentation masks as well as plane parameters for planes in the image, so that we can obtain the normal of planes to judge whether they are perpendicular or not. A pair of regions are *positive* if they are on perpendicular planes, and *negative* if they are not on perpendicular planes. Second row of Figure 2 is an example.

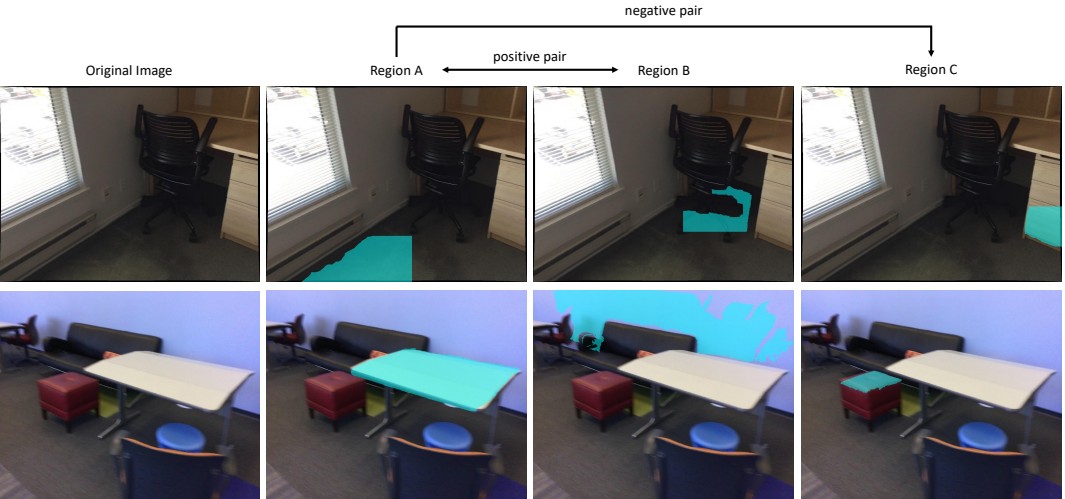

Figure 2: **Example images for probing *scene geometry*.** The first row shows a sample annotation for the *same plane*, and the second row is a sample annotation for *perpendicular plane*. Here, and in the following figures, (A, B) are a positive pair, while (A, C) are negative. The images are from the ScanNetv2 dataset (Dai et al., 2017) with annotations for planes from (Liu et al., 2019). In the first row, the first piece of floor (A) is on the same plane as the second piece of floor (B), but is not on the same plane as the surface of the drawers (C). In the second row, the table top (A) is perpendicular to the wall (B), but is not perpendicular to the stool top (C).

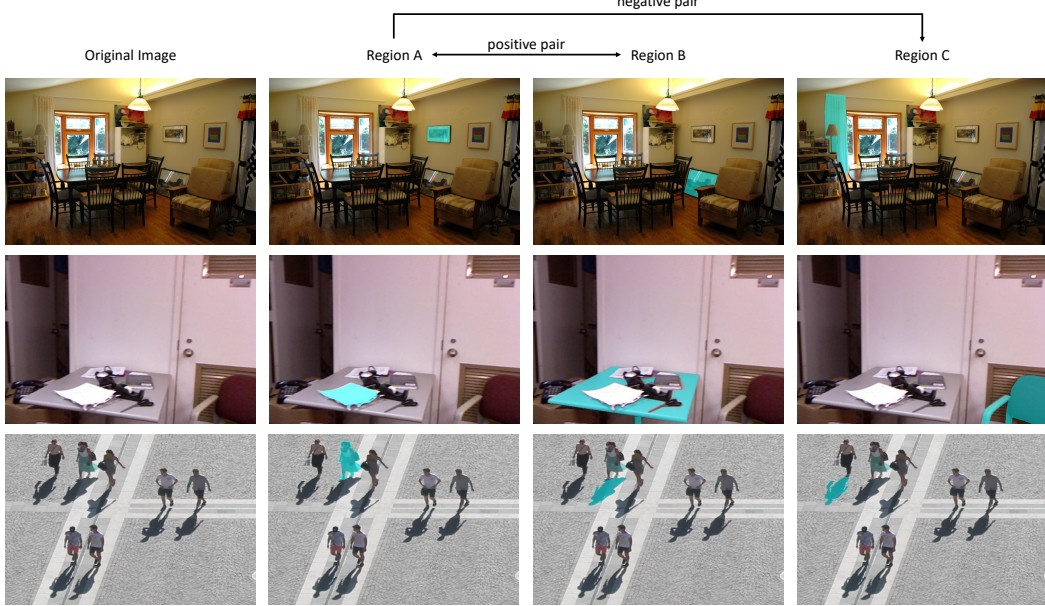

Figure 3: **Example images for probing *material, support relation and shadow*.** The first row is for *material*, the second row for *support relation*, and the third row for *shadow*. First row: the material images are from the DMS dataset (Upchurch & Niu, 2022). The paintings are both covered with glass (A and B) whereas the curtain (C) is made of fabric. Second row: the support relation images are from the NYUv2 dataset (Silberman et al., 2012). The paper (A) is supported by the table (B), but it is not supported by the chair (C). Third row: the shadow images are from the SOBA dataset (Wang et al., 2020). The person (A) has the shadow (B), not the shadow (C).

**Material.** We adopt the recent DMS dataset (Upchurch & Niu, 2022) to study the material property, which provides dense annotations of material category for each pixel in the images. Therefore, we can get the mask of each material via grouping pixels with the same material label together. In

total, there are 46 pre-defined material categories. Regions are obtained by splitting the mask of each material into different connected components, *i.e.,* they are simply groups with same material labels, yet not connected. A pair of regions are *positive* if they are of the same material category, and *negative* if they are of different material categories. First row of Figure 3 is an example.

**Support Relation.** We use the NYUv2 dataset (Silberman et al., 2012) to probe the support relation. Segmentation annotations for different regions (objects or surfaces) are provided, as well as their support relations. Support relation here means an object is physically supported by another object, *i.e.,* the second object will undertake the force to enable the first object to stay at its position. Regions are directly obtained via the segmentation annotations. A pair of regions are *positive* if the first region is supported by the second region, and *negative* if the first region is not supported by the second region. Second row of Figure 3 is an example.

**Shadow.** We use the SOBA dataset (Wang et al., 2020; 2021) to study the shadows which depend on the lighting of the scene. Segmentation masks for each object and shadow, as well as their associations are provided in the dataset annotations. Regions are directly obtained from the annotated object and shadow masks. In a region pair, there is one object mask and one shadow mask. A pair of regions are *positive* if the shadow mask is the shadow of the object, and *negative* if the shadow mask is the shadow of another object. Third row of Figure 3 is an example.

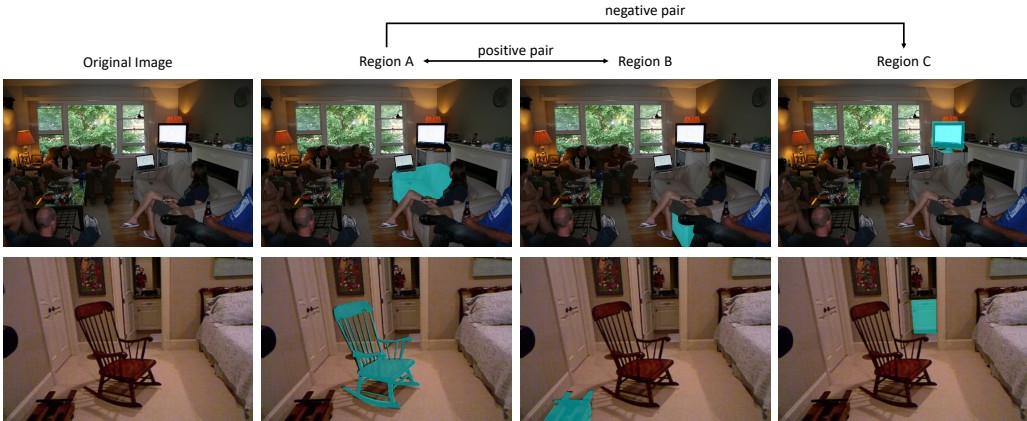

Figure 4: **Example images for probing *viewpoint-dependent properties (occlusion & depth)*.** The first row is for *occlusion* and the second row is for *depth*. First row: the occlusion images are from the Separated COCO dataset (Zhan et al., 2022). The sofa (*A*) and the sofa (*B*) are part of the same object, whilst the monitor (*C*) is not part of the sofa. Second row: the depth images are from the NYUv2 dataset (Silberman et al., 2012). The chair (*A*) is farther away than the object on the floor (*B*), but it is closer than the cupboard (*C*).

**Occlusion.** We use the Seperated COCO dataset (Zhan et al., 2022) to study the occlusion (object seperation) problem. Regions are different connected components of objects (and the object mask if it is not separated), *i.e.,* groups of connected pixels belonging to the same object. A pair of regions are *positive* if they are different components of the same object separated due to occlusion, and *negative* if they are not from the same object. First row of Figure 4 is an example.

**Depth.** We use the NYUv2 dataset (Silberman et al., 2012), that provides mask annotations for different objects and regions, together with depth for each pixel. A pair of regions are *positive* if the first region has a greater average depth than the second region, and *negative* if the first region has a less average depth than the second region. The average depth of a region is calculated via the average of depth value of each pixel the region contains. Second row of Figure 4 is an example.

### 3.3 PROPERTY PROBING

We aim to determine which Stable Diffusion features best represent different properties.

**Extracting Stable Diffusion Features.** Following DIFT (Tang et al., 2023), we add noise $\epsilon \sim \mathcal{N}(0, \mathbf{I})$ of time step $t \in [0, T]$ to the input image $x_0$'s latent representation $z_0$ encoded by the VAE

encoder:

$$z_t = \sqrt{\alpha_t} z_0 + (\sqrt{1 - \alpha_t})\epsilon \tag{1}$$

and then extract features from the immediate layers of a pre-trained diffusion model, $f_\theta(\cdot)$ after feeding $z_t$ and $t$ in $f_\theta$ ($f_\theta$ is a U-Net consisting of 4 downsampling layers and 4 upsampling layers):

$$F_{t,l} = f_{\theta_l}(z_t, t) \tag{2}$$

where $f_{\theta_l}$ is the $l$-th U-Net layer. In this way, we can get the representation of an image $F_{t,l}$ at time step $t$ and $l$-th U-Net layer for the probe. We upsample the obtained representation to the size of original image with bi-linear, then use the region mask to get a region-wise feature vector, by averaging the feature vectors of each pixel it contains, *i.e.,* average pooling.

$$v_{k,t,l} = \text{avgpool}(R_k \odot \text{upsample}(F_{t,l})) \tag{3}$$

where $v_{k,t,l}$ is the feature vector of the $k$-th region $R_k$. $\odot$ here is a per-pixel product of the region mask and the feature.

**Linear Probing.** After computing features from a diffusion model, we use a linear classifier (a linear SVM) to examine how well these features can be used to answer questions to each of the properties. Specifically, the input of the classifier is the difference or absolute difference between the feature vectors of Region *A* and Region *B*, *i.e.,* $v_A - v_B$ or $|v_A - v_B|$, and the output is a Yes/No answer to the question. Denoting the answer to the question as $Q$, then since the questions about *Same Plane*, *Perpendicular Plane*, *Material*, *Shadow* and *Occlusion* are symmetric relations, $Q(v_A, v_B) = Q(v_B, v_A)$. However, the questions about *Support Relation* and *Depth* are not symmetric. Thus, we use $|v_A - v_B|$ (a symmetric function) as input for the first group of questions, and $v_A - v_B$ (non-symmetric) for the rest of questions. We train the linear classifier on the train set via the positive/negative samples of region pairs for each property; do a grid search on the validation set to find (i) the optimal time step $t$, (ii) the U-Net layer $l$, and (iii) the SVM regularization parameter $C$; and evaluate the performance on the test set.

**Discussion.** Some of the current symmetric questions can be reformulated in a non-symmetric manner in order to obtain more information about the property. For example, the shadow question could be formulated as 'is region *A* the shadow of object *B*' rather than the (symmetric) 'are region *A* and region *B* in an object-shadow relationship'. The non-symmetric version requires the classifier to explicitly identify which region is the object, and which the shadow. Note, the protocol has been explained for diffusion models, but can **equally be applied to other pre-trained models**. In Section 4.3 and Section 4.4 we give results for its application to OpenCLIP (Radford et al., 2021; Ilharco et al., 2021), DINOv1 (Caron et al., 2021), DINOv2 (Oquab et al., 2023), and VQGAN (Esser et al., 2021).

## 4 EXPERIMENTS

### 4.1 IMPLEMENTATION DETAIL AND EVALUATION METRIC

**Implementation Details.** For each property, we sample the same number of positive / negative pairs, to maintain a balanced evaluation set. In terms of the linear SVM, we tune the penalty parameter $C$ on the val split to find the best $C$ for each property. Therefore, we are grid searching 3 parameters on the val set, namely, Stable Diffusion Timestep $t$ ranging from 0 to 1000, U-Net Layer $l$ covering the 4 downsampling and 4 upsampling layers, and the SVM penalty parameter $C$ ranging among $0.001, 0.01, 0.1, 1, 10, 100, 1000$. The linear SVM is solved using the *libsvm* library (Chang & Lin, 2011) with the SMO algorithm, to get the unique global optimal solution. Please refer to the supplementary for more implementation details.

**Evaluation Metric.** All protocols are binary classification, therefore, we use ROC Area Under the Curve (AUC Score) to evaluate the performance of the linear classifier, as it is not sensitive to different decision thresholds.

### 4.2 RESULTS FOR STABLE DIFFUSION

The results for grid search are shown in Table 2. For Stable Diffusion U-Net Layer, $D_l$ means the $l$-th layer of the U-Net decoder, *i.e.,* upsampling layer, from outside to inside, and we provide

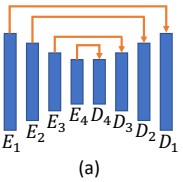 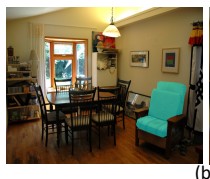 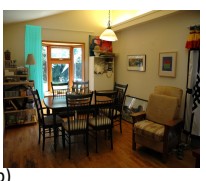 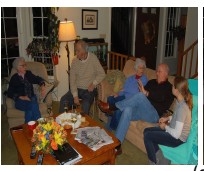 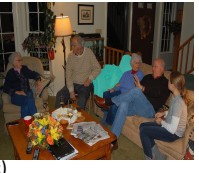

(a)                          (b)                                    (c)

Figure 5: **(a) Nomenclature for the U-Net Layers.** We probe 4 downsampling encoder layers $E_1$-$E_4$ and 4 upsampling decoder layers $D_1$-$D_4$ of the Stable Diffusion U-Net. **(b) A prediction failure for *Material*.** In this example the model does not predict that the two regions are made of the same material (fabric). **(c) A prediction failure for *Occlusion*.** In this example the model does not predict that the two regions belong to the same object (the sofa).

an illustration of the layers in Figure 5(a). We can draw 3 observations: First, it can be observed that the best time step for different properties are different, while the best U-Net layer is always in the decoder rather than the encoder. Further explorations using Stable Diffusion features for downstream tasks could thus start from the U-Net decoder layers, especially $D_3$ and $D_2$. Second, for *Material*, it is more about low-level features so we can observe that the best layer ($D_2$) is closer to the output side, while for the rest of the properties that require reasoning about the whole image, a more global feature is needed ($D_3$). Third, in terms of the performance on the test set, we find that Stable Diffusion can understand very well about scene geometry, support relations, shadows, and depth, but it is less performant at predicting material and occlusion. Examples of its failure are shown in Figure 5(b)(c). As noted in (Zhan et al., 2022) and (Kirillov et al., 2023), grouping all separated parts of an object due to occlusion remains challenging even for state-of-the-art detection and segmentation models.

Table 2: **SVM grid search results.** For each property, we train the linear SVM on the training set and grid search the best combination of time step, layer, and $C$ on the validation set. The ROC AUC score is reported on the test set using the selected combination.

| Property | Time Step | Layer | $C$ | Val AUC |
|---|---|---|---|---|
| Same Plane | 334 | $D_3$ | 1 | 97.2 |
| Perpendicular Plane | 126 | $D_3$ | 1 | 85.4 |
| Material | 339 | $D_2$ | 1 | 81.2 |
| Support Relation | 64 | $D_3$ | 1 | 95.2 |
| Shadow | 303 | $D_3$ | 1 | 96.1 |
| Occlusion | 181 | $D_3$ | 0.1 | 83.2 |
| Depth | 157 | $D_3$ | 1 | 99.5 |

### 4.3 RESULTS FOR OTHER FEATURES TRAINED AT LARGE SCALE

We have also applied our protocol to other models pre-trained on large scale image datasets, including OpenCLIP (Radford et al., 2021; Ilharco et al., 2021) pre-trained on LAION dataset (Schuhmann et al., 2022), DINOv1 (Caron et al., 2021) pre-trained on ImageNet dataset (Deng et al., 2009), DINOv2 (Oquab et al., 2023) pre-trained on LVD-142M dataset (Oquab et al., 2023), and VQGAN Esser et al. (2021) pre-trained on ImageNet dataset (Deng et al., 2009). We use the best pre-trained checkpoints available on their official GitHub – ViT-B for DINOv1, ViT-G for OpenCLIP and DINOv2, and the 48-layer transformer checkpoint for VQGAN. Similar to Stable Diffusion, for each of these models, we conduct a grid search on the validation set in terms of the ViT/Transformer layer and $C$ for SVM, and use the best combination of parameters for evaluation on the test set.

Performance on both val and test set in AUC score is reported in Table 3 for the task of material and support. We also compare the performance of the best layer to that of the final layer. We can observe that for both properties and for all four models, the test performance is lower than that of Stable Diffusion by a margin. Results of other tasks are in the supplementary (again Stable Diffusion is superior).

Table 3: **Performance of different layers for state-of-the-art pre-trained models.** We train the linear SVM on the training set, and grid search the best combination of ViT layer and $C$ on the validation set for the material and support relation properties. The ROC AUC is reported on the test set using the selected combination. The test performance of the selected layer may be better than the last layer, but is still considerably lower than that of the Stable Diffusion feature.

| Layer | Split | Material | | | | | Support Relation | | | | |
|---|---|---|---|---|---|---|---|---|---|---|---|
| | | OpenCLIP | DINO v1 | DINO v2 | VQGAN | Stable Diffusion | OpenCLIP | DINO v1 | DINO v2 | VQGAN | Stable Diffusion |
| Last | Val | 58.5 | 55.3 | 59.3 | 56.8 | - | 82.2 | 82.4 | 82.9 | 70.9 | - |
| Best | Val | 64.1 | 64.0 | 63.4 | 63.4 | 81.2 | 85.4 | 82.9 | 86.9 | 83.4 | 95.2 |
| Last | Test | 60.4 | 62.1 | 63.8 | 53.3 | - | 84.7 | 84.3 | 88.3 | 71.5 | - |
| Best | Test | 64.3 | 65.3 | 63.2 | 62.6 | 79.4 | 86.4 | 84.3 | 88.5 | 84.3 | 94.4 |

Table 4: **Performance of Stable Diffusion features compared to state-of-the-art self-supervised features.** For each property, we use the best time step, layer and $C$ found in the grid search in Table 2 for Stable Diffusion, and the best layer and $C$ found in the grid search for other features. The performance is the ROC AUC on the test set, and 'Random' means a random classifier.

| Property | Random | OpenCLIP | DINOv1 | DINOv2 | VQGAN | Stable Diffusion |
|---|---|---|---|---|---|---|
| Same Plane | 50 | 84.3 | 82.9 | 84.5 | 78.4 | **95.0** |
| Perpendicular Plane | 50 | 61.1 | 58.6 | 66.2 | 54.9 | **83.9** |
| Material | 50 | 64.3 | 65.3 | 63.2 | 62.6 | **79.4** |
| Support Relation | 50 | 86.4 | 84.3 | 88.5 | 84.3 | **94.4** |
| Shadow | 50 | 92.0 | 86.9 | 87.0 | 85.9 | **94.5** |
| Occlusion | 50 | 65.6 | 62.0 | 67.1 | 60.4 | **75.6** |
| Depth | 50 | 97.7 | 94.4 | 98.4 | 90.5 | **99.3** |

## 4.4 COMPARISON OF DIFFERENT FEATURES TRAINED AT LARGE SCALE

We compare the state-of-the-art pre-trained large-scale models' representations on various downstream tasks in Table 4. It can be observed that the Stable Diffusion representation outperforms all the other models for all properties and achieves the best performance, indicating the potential of utilizing Stable Diffusion representation for different downstream tasks with the optimal time steps and layers found in Section 4.2.

## 5 DISCUSSION AND FUTURE WORK

In this paper, we have developed a protocol to examine whether the Stable Diffusion model has explicit feature representations for different properties of the 3D scene. Our method, using off-the-shelf annotated image datasets and a linear probe of the Stable Diffusion representation, can also be applied to other models pre-trained on large scale image datasets, like CLIP and DINO.

It is interesting to find that different time steps and layers of Stable Diffusion representations can handle several different physical properties at a state-of-the-art performance, indicating the potential of utilising the Stable Diffusion model for different downstream tasks. However, some properties such as material and occlusion as evaluated in (Upchurch & Niu, 2022) and (Zhan et al., 2022) are still challenging for large scale pre-trained models such as Stable Diffusion, DINO, and CLIP. Though occlusion is a challenge even for the powerful Segment Anything Model (SAM) (Kirillov et al., 2023), where it is noted that the model 'hallucinates small disconnected components at times'.

This paper has given some insight into answering the question: 'What does Stable Diffusion know about the 3D scene?'. Of course, there are more properties that could be investigated in the manner proposed here. For example, contact relation (Fouhey et al., 2016) and object orientation (Xiang et al., 2018), as well as the more nuanced non-symmetric formulations of the current questions. Another direction would be to use the pixel-wise supervision method of (Bhattad et al., 2023) to search for features that can predict depth maps, normal maps, *etc*. We leave these for the future.

**Reproducibility Statement.** We discuss the datasets we used in Section 3.2 of the main paper, provide implementation details in Section 4.1 of the main paper, and more implementation details in the supplementary to ensure reproducibility. All datasets and code will be publicly released.

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

SUPPLEMENTARY

## A    MORE IMPLEMENTATION DETAILS

**Extracting Stable Diffusion Features.**    Following DIFT (Tang et al., 2023), when we extract Stable Diffusion features, we add a different random noise 8 times and then take the average of the generated features. We use an empty prompt '' as the text prompt.

**Train/Val Partition.**    For the partition of train/val split, we select the train & val images from different scenes for the NYUv2 (Silberman et al., 2012) and ScanNetv2 (Dai et al., 2017) dataset.

**Sampling of Images.**    For the train/val/test splits, if the number of images used is less than the original number of images in the datasets, we randomly sample our train/val/test images from the original datasets.

**Sampling of Positive/Negative Pairs.**    For each property, we try to obtain as many positive/negative region pairs as possible in every image. For each image, if the number of possible negative pairs is larger than the number of possible positive pairs, we randomly sample from the negative pairs to obtain an equal number of negative and positive pairs, and vice versa. In this way, we keep a balanced sampling of positive and negative pairs for the binary linear classifier. As can be observed in Table 1, the number of train/val pairs for different properties are different, although we keep the same number of train/val images for different properties. This is because for different properties the availability of positive/negative pairs are different. For *depth*, we select a pair only if the average depth of one region is 1.2 times greater than the other because it is even challenging for humans to judge the depth order of two regions below this threshold. For *perpendicular plane*, taking the potential annotation errors into account, we select a pair as perpendicular if the angle between their normal vectors is greater than $85°$ and smaller than $95°$, and select a pair as not perpendicular if the angle between their normal vectors is smaller than $60°$ or greater than $120°$.

**Region Filtering.**    When selecting the regions, we filter out the small regions, *e.g.,* regions smaller than 1000 pixels, because regions that are too small are challenging even for humans to annotate.

**Image Filtering.**    As there are some noisy annotations in the (Liu et al., 2019) dataset, we manually filter the images whose annotations are inaccurate.

**Linear SVM.**    The feature vectors are L2-normalised before inputting into the linear SVM. The binary decision of the SVM is given by $sign(w^T v + b)$, where $v$ is the input vector to SVM:

$$v = |v_A - v_B| \tag{4}$$

for the *Same Plane*, *Perpendicular Plane*, *Material*, *Shadow* and *Occlusion* questions, and

$$v = v_A - v_B \tag{5}$$

for the *Support Relation* and *Depth* questions.

**Extension of Separated COCO.**    To study the occlusion problem, we utilise the Separated COCO dataset (Zhan et al., 2022). The original dataset only collects separated objects due to occlusion in the COCO 2017 val split, we further extend it to the COCO 2017 train split for more data using the same method as in (Zhan et al., 2022).

## B    ANALYSIS OF STABLE DIFFUSION GENERATED IMAGES

As Figure 1 shows, our motivation for the paper is that we observe that Stable Diffusion correctly predicts different physical properties of the scene. The reason why we do not study the generated images directly is that there are no annotations available on different properties for these synthetic images, so it is expensive to get quantitative results. But in this section, we provide more qualitative examples and analysis of Stable Diffusion generated images in terms of different physical properties. The observations match our findings in the main paper – Stable Diffusion 'knows' about a number of physical properties including scene geometry, material, support relations, shadows, occlusion and depth, but may fail in some cases in terms of material and occlusion.

We show examples for: **Scene Geometry** in Figure 6; **Material**, **Support Relations**, and **Shadows** in Figure 7; and **Occlusion** and **Depth** in Figure 8.

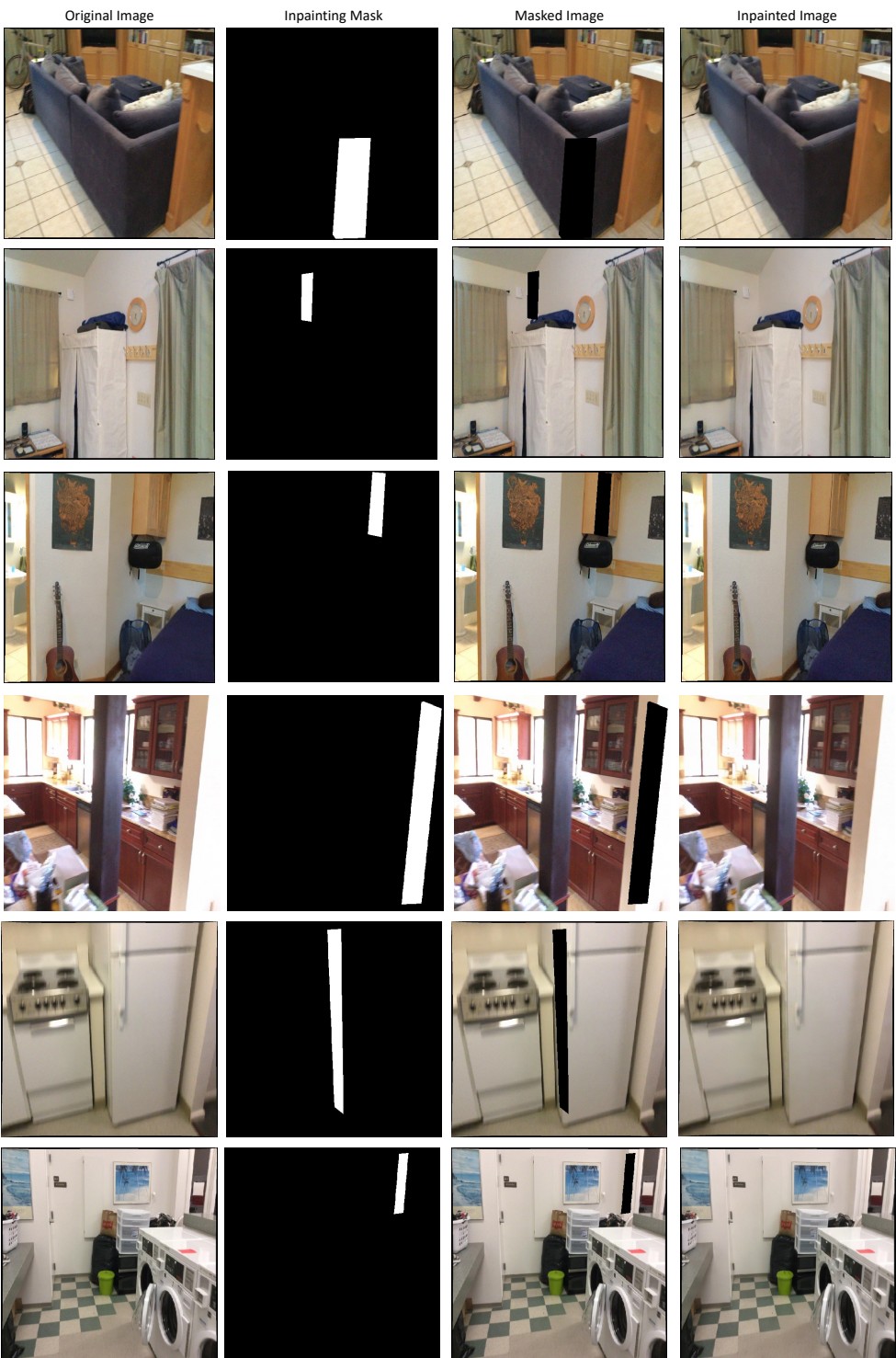

Figure 6: **Stable Diffusion generated images testing** *scene geometry* **prediction.** Here and for the following figures, the model is tasked with inpainting the masked region of the real images. Stable Diffusion 'knows' about *same plane* and *perpendicular plane* relations in the generation. When the intersection of two sofa planes (first row), two walls (second and sixth row), two cabinet planes (third row), two pillar planes (fourth row) or two fridge planes (fifth row) is masked out, Stable Diffusion is able to generate the two perpendicular planes at the corner based on the unmasked parts of the planes.

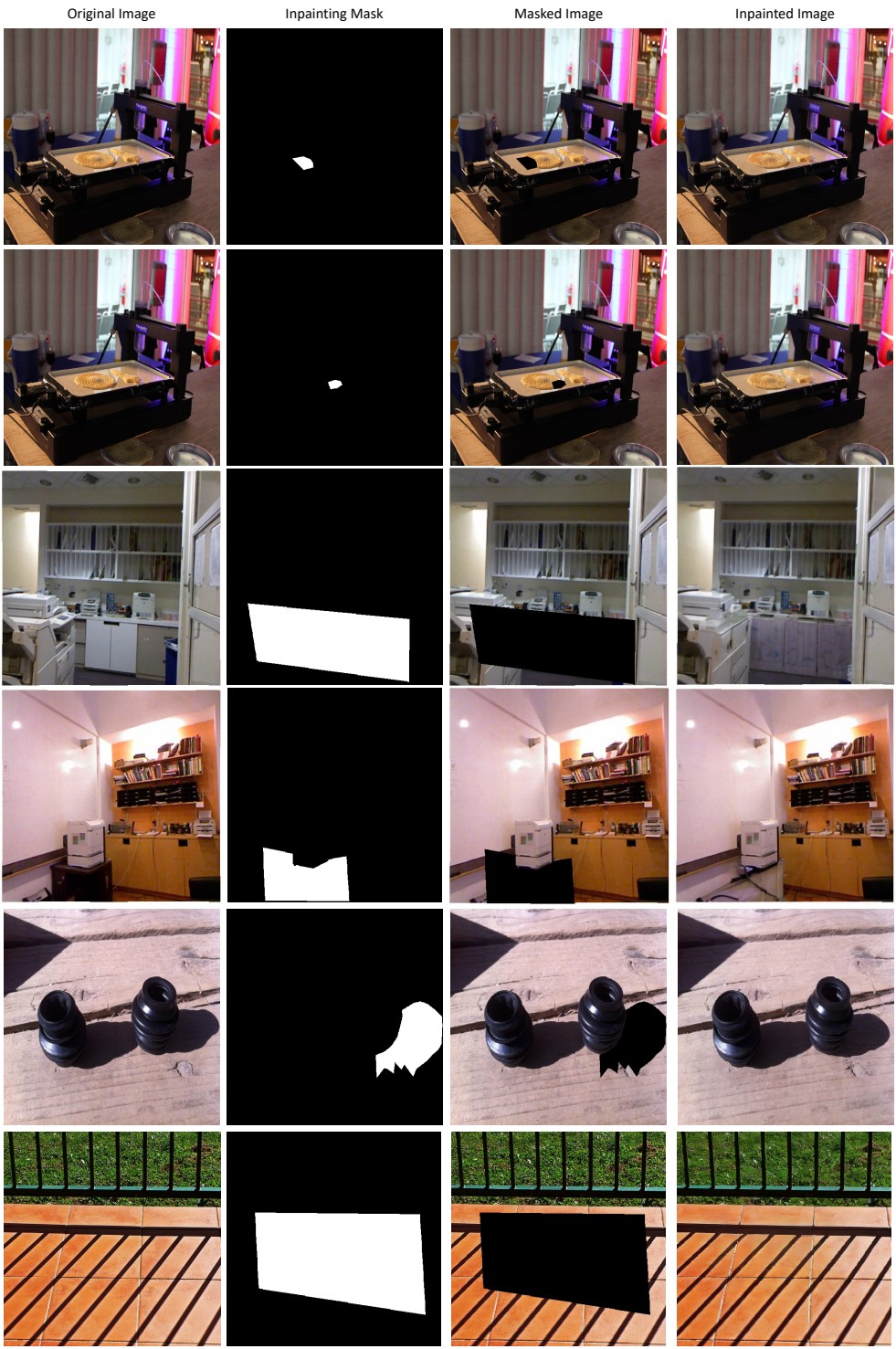

Figure 7: **Stable Diffusion generated images testing *material*, *support relation* and *shadow* pre-diction.** Stable Diffusion 'knows' about *support relations* and *shadows* in the generation, but may fail sometimes for *material*. Rows 1-2: Material; Rows 3-4: Support Relation; Rows 5-6: Shadow. In the first row, the model distinguishes the two different materials clearly and there is clear boundary between the generated pancake and plate; while in the second row, the model fails to distinguish the two different materials clearly, generating a mixed boundary. In the third row and fourth rows, the model does inpaint the supporting object for the stuff on the table and the machine. In the fifth and sixth rows, the model manages to inpaint the shadow correctly. Better to zoom in for more details.

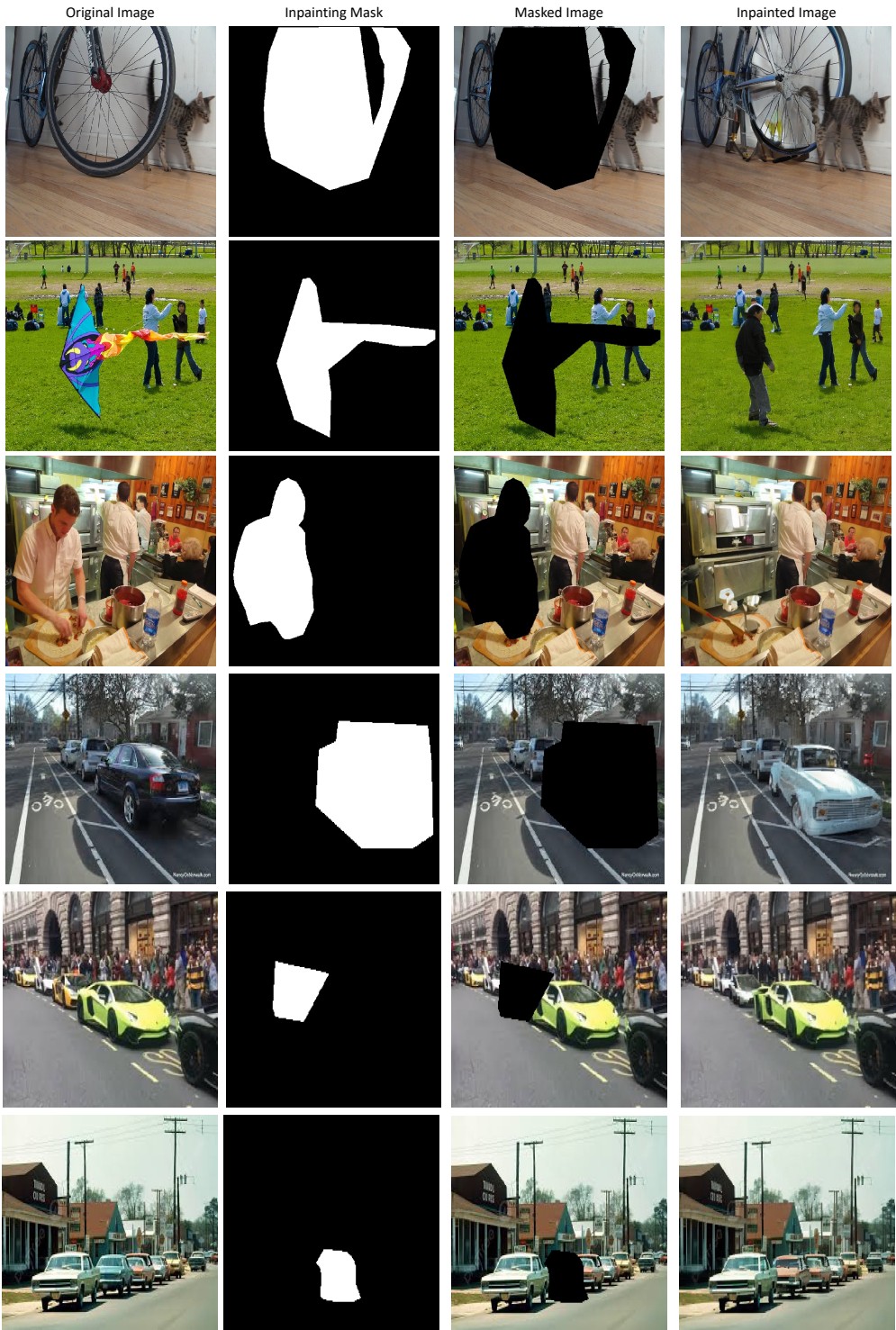

Figure 8: **Stable Diffusion generated images testing *occlusion* and *depth* prediction.** Stable Diffusion 'knows' about *depth* in the generation, but may fail sometimes for *occlusion*. Rows 1-3: Occlusion; Rows 4-6: Depth. In Row 1, the model fails to connect the tail with the cat body and generates a new tail for the cat, while in Row 2, the model successfully connects the separated people and generates their whole body, and in Row 3, the separated parts of oven are connected to generate the entire oven. In Rows 4-6, the model correctly generates a car of the proper size based on depth. The generated car is larger if it is closer, and smaller if it is farther away.

## C ADDITIONAL RESULTS FOR OTHER FEATURES TRAINED AT LARGE SCALE

As mentioned in Section 4.3 of the main paper, we have conducted grid search for all the other large pre-trained models, including OpenCLIP, DINOv1, DINOv2 and VQGAN for all tasks. Tables in this section provide results for the Same Plane (Table 5), Perpendicular Plane (Table 6), Shadow (Table 7), Occlusion (Table 8) and Depth (Table 9) tasks for these models. It can be observed that for all tasks, the test performance of each model is improved if we take the best combination of layer and $C$ on the val split, but the performance is still lower than Stable Diffusion.

Table 5: **Performance of different layers for state-of-the-art pre-trained models for the Same Plane task.**

| Layer | Split | Same Plane | | | | |
| --- | --- | --- | --- | --- | --- | --- |
| | | OpenCLIP | DINO v1 | DINO v2 | VQGAN | Stable Diffusion |
| Last | Val | 72.7 | 74.9 | 80.9 | 65.2 | - |
| Best | Val | 84.4 | 81.7 | 82.1 | 77.5 | 97.2 |
| Last | Test | 74.6 | 79.3 | 86.0 | 65.4 | - |
| Best | Test | 84.3 | 82.9 | 84.5 | 78.4 | 95.0 |

Table 6: **Performance of different layers for state-of-the-art pre-trained models for the Perpendicular Plane task.**

| Layer | Split | Perpendicular Plane | | | | |
| --- | --- | --- | --- | --- | --- | --- |
| | | OpenCLIP | DINO v1 | DINO v2 | VQGAN | Stable Diffusion |
| Last | Val | 54.9 | 54.1 | 62.8 | 54.6 | - |
| Best | Val | 62.6 | 58.9 | 68.5 | 61.3 | 85.4 |
| Last | Test | 55.5 | 59.8 | 63.4 | 50.2 | - |
| Best | Test | 61.1 | 58.6 | 66.2 | 54.9 | 83.9 |

Table 7: **Performance of different layers for state-of-the-art pre-trained models for the Shadow task.**

| Layer | Split | Shadow | | | | |
| --- | --- | --- | --- | --- | --- | --- |
| | | OpenCLIP | DINO v1 | DINO v2 | VQGAN | Stable Diffusion |
| Last | Val | 78.1 | 85.4 | 88.5 | 50.0 | - |
| Best | Val | 93.9 | 88.8 | 90.2 | 86.0 | 96.1 |
| Last | Test | 75.5 | 84.3 | 86.8 | 50.8 | - |
| Best | Test | 92.0 | 86.9 | 87.0 | 85.9 | 94.5 |

Table 8: **Performance of different layers for state-of-the-art pre-trained models for the Occlusion task.**

| Layer | Split | Occlusion | | | | |
|-------|-------|-----------|--------|--------|-------|------------------|
| | | OpenCLIP | DINO v1 | DINO v2 | VQGAN | Stable Diffusion |
| Last | Val | 61.5 | 65.3 | 65.8 | 49.7 | - |
| Best | Val | 74.0 | 71.3 | 70.3 | 72.5 | 83.2 |
| Last | Test | 63.8 | 60.0 | 67.9 | 53.9 | - |
| Best | Test | 65.6 | 62.0 | 67.1 | 60.4 | 75.6 |

Table 9: **Performance of different layers for state-of-the-art pre-trained models for the Depth task.**

| Layer | Split | Depth | | | | |
|-------|-------|-------|--------|--------|-------|------------------|
| | | OpenCLIP | DINO v1 | DINO v2 | VQGAN | Stable Diffusion |
| Last | Val | 96.8 | 94.4 | 97.5 | 79.4 | - |
| Best | Val | 98.4 | 95.5 | 98.4 | 90.9 | 99.5 |
| Last | Test | 95.5 | 93.7 | 98.0 | 73.8 | - |
| Best | Test | 97.7 | 94.4 | 98.4 | 90.5 | 99.3 |

## D    VISUALISATION OF STABLE DIFFUSION FEATURE REPRESENTATIONS

In Figure 9 we visualise the vectors representing the positive/negative pairs in the Depth and Material tasks using t-SNE. It is obvious that the vectors are easier to be separated for the Depth task than the Material task, which confirms to the observation that we get a higher AUC when we apply linear SVM to the depth task but lower AUC when we apply it to the material task. In the future, more efforts should be put into training the Stable Diffusion model to have a better understanding of Material and Occlusion, *e.g.*, explicitly incorporate these tasks into training.

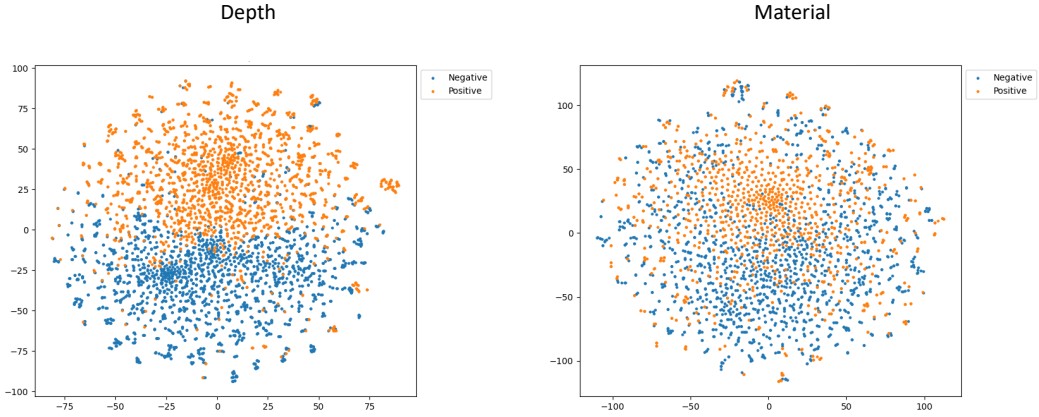

Figure 9: **t-SNE Visualisation of Stable Diffusion feature space for the *Depth* and *Material* tasks.** It can be observed that the vectors for the depth task are more easy to separate than the material, which confirms to the observation that we get a higher AUC when we apply linear SVM to the depth task but lower AUC when we apply it to the material task.

