# OpenReview forum: "What Does Stable Diffusion Know about the 3D Scene?"
_ICLR.cc/2024/Conference — Submitted to ICLR 2024_

### Official Review · Reviewer_n8LA · 2023-10-26

**Soundness:** 3 good
**Presentation:** 3 good
**Contribution:** 2 fair
**Rating:** 5
**Confidence:** 4

**Summary:**

This paper shows diffusion models have better scene understanding capabilities than DINO and CLIP. This is demonstrated by training a linear classifier on the output of specific layers of these pre-trained models to learn to classify whether the relationships between different regions in a 3D scene can be distinguished.

**Strengths:**

- The paper is easy to read, and generally well written.
- This paper examines the ability of a diffusion model to understand different physical properties of the 3D scene, including scene geometry, material, support relations, shadows, occlusion, and depth.
- The paper also shows that stable diffusion models can be used as a feature extractor for a wider range of downstream tasks.

**Weaknesses:**

1) The paper does not compare the proposed method to other generative models, such as StyleGAN[1] and VQGAN[2], and other diffusion models such as DDPM[3] and GLIDE[4]. This would help to provide more insights on whether generative model generally gains good performance on 3D scene understanding or only diffusion models. This is a significant weakness, as it is difficult to assess the significance of the power of the diffusion model without knowing how it compares to other methods.

2) Diffusion models have been used in different tasks for 3D data [5][6] and demonstrated promising results in different tasks. There are also other different important aspects of 3D scene understanding, eg. scene categories, attributes, and structures.[7] How does the author choose these 7 properties, are these 7 properties enough to “understand” the 3D scene, what problems this paper is trying to solve by experimenting on these properties, and the challenges that lie on them?

3) This paper lacks of a more complete ablation study, which could provide more insight. For example, how does the number of time steps or layers affect the task? Is it sensitive to these parameters?

4) The paper title is not really what the paper is about. This paper is more on the interpretability of a trained model, no matter it is part of the DINO or CLIP models, or stable diffusion models.

Summary Of The Review:

This is an empirical paper and the paper could be further improved by including a more comprehensive experiments and ablation study, e.g. comparison to other state-of-the-art methods, in particular other generative models and other diffusion models. This would make the paper stronger and more informative for readers.

References:
[1] A Style-Based Generator Architecture for Generative Adversarial Networks
[2] Taming Transformers for High-Resolution Image Synthesis
[3] Denoising Diffusion Probabilistic Models
[4] GLIDE: Towards Photorealistic Image Generation and Editing with Text-Guided Diffusion Models
[5] Diffusion Probabilistic Models for Scene-Scale 3D Categorical Data
[6] Diffusion-based Generation, Optimization, and Planning in 3D Scenes
[7] Basic level scene understanding: from labels to structure and beyond

**Questions:**

See my questions in 2) and 3) above.

---

> ### Author Response · Authors · 2023-11-21
> **Rebuttal by Authors**
>
> Thanks for valuable comments! We have addressed them below. Hope to hear back from you if you have further questions!
>
> W1 (Evaluation of other generative models): See Reviewer G3yb W3. As mentioned in the overall comments, our objective was to propose a protocol and use this to evaluate a set of models to compare to Stable Diffusion, but not to comprehensively evaluate all models. We have added an evaluation for a second generative model, VQGAN, and updated the results in our revision, showing that Stable Diffusion still has a superior performance (Table 4). DINO and CLIP models are compared because they are strong self-supervised features and are used extensively for scene understanding tasks.
>
> W2 (Choice of 7 properties): The 7 properties we choose are important in terms of understanding the 3D physical scene. As mentioned in the discussion and future work section of the paper, we aim to provide a protocol; other properties can be studied in the same way as introduced in the paper. The discovery of the paper indicates that although Stable Diffusion has been trained for generative tasks, features extracted from the model can also be applied in a variety of downstream tasks including material segmentation, shadow association, support relation reasoning, 3D reconstruction, depth estimation, occlusion handling (separated parts association), which are the important challenges in terms of physical scene understanding. The paper provides the best time step and layer for different tasks of Stable Diffusion features.
>
> W3 (Different time steps and layers): As mentioned in Section 4.1 of the paper, we have conducted a grid search in terms of different time steps and layers to find the best time step and layer for each task. We can observe that usually D3 and D2 perform best and the best time step is usually between 0-400 as later time steps might have added too much noise, as in Section 4.2 and Table 2.
>
> W4 (Paper title): We want to probe how well different models understand different properties about the 3D scene, mainly for Stable Diffusion, so the title reflects what the paper is about. The paper is answering the question ‘What Does Stable Diffusion Know about the 3D Scene?’

---

### Official Review · Reviewer_ssWi · 2023-10-30

**Soundness:** 2 fair
**Presentation:** 3 good
**Contribution:** 3 good
**Rating:** 5
**Confidence:** 3

**Summary:**

To better understand and measure the 3D scene modeling capability of the 2D stable diffusion, the author has designed an interesting evaluation protocol by probing for explicit features that represent physical properties. A linear SVM is leveraged to examine how well these these averaged patch-wise features from SD can be used to answer questions on physical properties. This work has done extensive experiments on different 3D properties including scene geometry, scene material, support relations, lighting, and view dependent measures, and throw insights on SD's superior modeling of a few physical attributes, but less performant for occlusion.

**Strengths:**

[**originality**]
The author proposes to use the capability of serving as robust physical properties classifier as standards to evaluate the SD's feature on physical attributes modeling, which is novel and makes sense. The author further uses domain-specific datasets for each individual attribute, which disentangles different physical property in a unified framework.

[**significance**]
While there are several previous works touch the field of interpreting pre-trained generative model like SD, quantitative evaluation is still missing, and this work has partially addressed this issue. The proposed framework, including the designed the binary questions are fully automatic using existing labels from available datasets, this can save human labeling efforts, and make the pipeline applicable to further T2I model development.

[**quality** & **clarity**]
The paper is written in a structured way, while complete experiments have been performed in various settings with a clear conclusion.

**Weaknesses:**

- I am a little skeptical of the key conclusion presented in Table 2, does that simply mean Layer 3 handles all physical properties while other layers fail? This might be true for early layers, but I am not quite sure about D4. A related concern is related to the feature averaging, maybe the conclusion is wrongly drawn since we didn't choose the suitable patch size, a bigger or smaller patch size might lead to a different conclusion. For designing such average feature, would the feature be more informative if we do average pooling to get 2x2 grid and then flatten the feature, in this way, the physical spatial relationship is also preserved. The author might want to have more ablations on designing the feature.
- The author's experiments are limited to feature quality evaluation, but there are no explicit visualization of the latent feature, and there is no example how such evaluation can benefit and help us design the next-generation T2I model, or Text-to-3D models by improving over the feature space.
- For the Question 6 design on occlusion, I feel it is not appropriate since there are two concepts: 1. belonging to same object; 2. whether there are occlusion. Before we evaluate occlusion, we need to first know how the classifier will work for parts belonging to same object but without occlusion.

**Questions:**

- These datasets with GT labels might serve as domain-specific experts, in order to fully capture the learnt image distribution of SD, how do we select these training/test images?

---

> ### Author Response · Authors · 2023-11-21
> **Rebuttal by Authors**
>
> Thanks for valuable comments! We have addressed them below. Hope to hear back from you if you have further questions!
>
> W1 (Conclusion in Table 2): It means that for most properties the best layer is Layer 3 but at different time steps. Other layers are not failing, simply less performant than Layer 3 at the certain time step. For example, for the depth task, the best performance (AUC) of each layer over all time steps and C is E1:85.9, E2:93.9, E3:98.6, E4:98.2, D4:98.7, D3:99.5, D2:98.5, D1:95.0. It is reasonable that D4 is not performing the best, as it is the first layer of denoising decoding and might still contain some noise. For feature averaging, as mentioned in the paper, we ‘use the region mask to get a region-wise feature vector, by averaging the feature vectors of each pixel it contains’, so there is nothing concerning the patch size. It is different from the traditional ‘average pooling’ as our regions are not in regular shapes, e.g., a rectangle.
>
> W2 (Visualisation of latent feature): We have updated the t-SNE visualization of the feature vectors in our revision (Section D in the supplementary), where we can observe that the better performance the SVM is getting, the better distribution these vectors are getting in the embedding space, i.e., the positive vectors and negative vectors are distributed apart from each other. In the conclusion of the paper, we conclude that current Stable Diffusion model could understand a variety of physical properties while less performant at the occlusion and material tasks, which indicates the future research could focus on improving these properties of Stable Diffusion by explicitly incorporating these tasks into training.
>
> W3 (Design of occlusion question): We understand the reviewer’s point about asking the question first about whether two parts belong to the same object. We will clarify this in the discussion. It is partially covered already by the co-planar question. In the current question we are following the dataset and challenge proposed by [1], since we want to investigate the view-point dependence of occlusion here (rather than grouping `parts’ into an object, which is a segmentation problem). Segmentation has already been studied for Stable Diffusion.
>
> Q1 (Selection of train/test images): As mentioned in Section A of the supplementary, the training/testing images are randomly sampled from the original datasets. We are testing on these datasets because they have ground truth annotations for different physical properties.
>
> References:
> [1] A Tri-Layer Plugin to Improve Occluded Detection. Guanqi Zhan, Weidi Xie, Andrew Zisserman. BMVC 2022

---

> > ### Comment · Reviewer_ssWi · 2023-11-23
> >
> > Thanks for the authors' reply, several of my concerns have been resolved. Regarding the significance, how such evaluation can benefit and help us design the next-generation T2I model, or Text-to-3D models by improving over the feature space?

---

> > > ### Author Response · Authors · 2023-11-23
> > > **Reply to Reviewer ssWi**
> > >
> > > Thanks so much for your reply. We are focusing on exploring whether the pre-trained generative Stable Diffusion model could understand the physical properties of the scene, rather than design a better generative model for image/3D generation. Through the thorough experiments, we demonstrate the generative diffusion model could outperform other large pre-trained models on various tasks such as scene geometry, material, shadow, support relation, occlusion and depth. This is significant for the follow-ups to explore the large generative models for understanding tasks in our community.
> > > Furthermore, we found that the features of Stable Diffusion model perform worse on occlusion and materials. We believe this is a reasonable analysis on why the existing Stable Diffusion model can not simultaneously generate many objects in one scene with complex 3D layout, such as occlusion. We think the future training of Stable Diffusion can explicitly incorporate these task into training, for example, with classifier-guidance of occlusion and material, so that the feature space could also be improved for occlusion and material (Figure 9 in the supplementary).

---

### Official Review · Reviewer_G3yb · 2023-11-01

**Soundness:** 2 fair
**Presentation:** 2 fair
**Contribution:** 2 fair
**Rating:** 3
**Confidence:** 4

**Summary:**

The paper presents a study on the 3D awareness of stable diffusion in the setting of infilling uncertain areas. They focus on questions of perpendicular/same plane for geometry, surface material, support relation, lighting (shadows), occlusion and depth for viewpoint. They experiment across different datasets depending on the information provided within. To answer the binary question (yes/no) they train a classifier extracted from features within the U-Net of Stablediffusion, which they ablate on layer depending on each setting, where the authors find that a single generator layer is responsible for most decisions. They show a comparison against other methods DINO and CLIP and show that Stable Diffusion out performs in these reasoning questions.

**Strengths:**

- The benchmark setting is an interesting way of testing future image completion methods. The author's test suite, although relatively small, can be easily applied, although the computational cost and fairness of the ablation of layers could be an issue depending on the architectures.
- Given the prominence of Stablde Diffusion, the analysis is an interesting way to quantify what it is aware of. There are currently very limited studies (as the authors highlight) exploring a number of different questions about spatial awareness of models.

**Weaknesses:**

- The paper oversells the approach at the beginning of the paper. It would be better to have the title/abstract/introduction define the constraint of in-filling, as this changes the problem significantly. In general, we see that in-filling is a significantly easier task than generating from scratch, which is more the common use case for Stablediffusion. Therefore, it would be better to constrain the arguments of the paper to be specific to this, as the benchmark would not generalise to unconstrained image generation. (This is why reduced Soundness and Presentation scores as the paper is misleading).
- The use of ScanNet is problematic for plane tests as the segmentation is quite incomplete. Therefore, in a large case the part you are infilling you can see the rest of the wall/floor/surface giving strong context clues. These tests provide limited insights.
- It would have been good to see other models evaluated in the same setting, even older approaches like GANs or VQ-VAE or StableDiffusion derivatives to provide contextual information. While DINO and CLIP provide insights, they are focused on a different task and probably explain the worse performance. A single focus to the paper has a limited setting for arguing a benchmark.
- It is unclear why they only ablate across the SVM C parameter and not the Gamma as well. While in a large amount of cases, this has limited effect, it should be justified ideally empirically.
- The authors argue a complete study in the beginning and then only evaluate DINO and Clip on Material and Support. The claim should either be consistent or the experiments completed.
- Missing human performance in the table to judge the difficulty of the task as the results are very high, implying the questions are quite easy.

**Questions:**

- It would be useful to understand why a single model for evaluation can justify a benchmark setting for future evaluation. Or if the authors weren't thinking of it as much as a benchmark and more of an analysis of StableDiffusion alone.
- Whether they tested human performance on the model? And how they came up with the groundtruth

---

> ### Author Response · Authors · 2023-11-21
> **Rebuttal by Authors**
>
> We suspect that the reviewer has misunderstood the paper as they stress inpainting as our goal. Our proposed protocol does not involve inpainting at all. We included one figure showing inpainting as a motivational example for the power of stable diffusion. We say in the text that this figure “is an indication that the diffusion network is 3D and physics aware”, and have now clarified in the figure caption of the revision that it is a motivational example. All the other figures do show the properties we evaluate on. However, given the reviewer’s apparent misunderstanding, it is difficult to respond to some of their questions.
>
> W2 (Use of ScanNet for plane test): As we are not studying the inpainting model, the masks in Figure 2 are only used to extract the Stable Diffusion feature of the specified region, and it is okay if the mask is smaller than the whole region (the mask can be any sub-region of the whole region).
>
> W3 (Evaluating other models): As mentioned in the overall comments, our objective was to propose a protocol and use this to evaluate a set of models to compare to Stable Diffusion, but not to comprehensively evaluate all models. We have added an evaluation for a second generative model, VQGAN, and updated the results in our revision, showing that Stable Diffusion still has a superior performance (Table 4). DINO and CLIP models are compared because they are strong self-supervised features and are used extensively for scene understanding tasks.
>
> W4 (Gamma in SVM): We have mentioned many times in the paper that we are using a linear SVM. This does not have a gamma parameter.
>
> W5 (Grid search for other models): See Reviewer oLXV W3. We have now updated results to have a full grid search for all the foundation models we evaluate on (Table 3,4,5,6,7,8,9 in the revision). Stable Diffusion still outperforms all of them.
>
> W6 (Questions are quite easy): As can be seen in Table 4, the performance of other large pre-trained models for these questions including DINO, CLIP and VQGAN is not high, indicating the questions are not easy. The performance of Stable Diffusion is high because Stable Diffusion has strong capability in terms of different physical properties.
>
> Q1 (Evaluation for other models): Complete evaluation of CLIP and DINO is provided in the revision, as well as an evaluation on VQGAN.
>
> Q2 (Ground truth for evaluation): On the Ground Truth (GT), as explained in the paper (Section 3.2) we are using standard image benchmark datasets where the GT for these properties is provided. As mentioned in Section B of our supplementary, “the reason why we do not study the generated images directly is there are no annotations about different properties for these synthetic images”. The quality of GT annotations for these real images datasets is of high quality, and they are all from well-established works.

---

### Official Review · Reviewer_oLXV · 2023-11-06

**Soundness:** 2 fair
**Presentation:** 4 excellent
**Contribution:** 2 fair
**Rating:** 5
**Confidence:** 3

**Summary:**

This paper proposes a method for testing whether the features from a pre-trained network can be used to classify certain geometric and physical relations between a pair of masked regions in an image. These relations can be about 3D structure, e.g. whether two regions are on the same plane. They can also be about physical properties, e.g. whether two regions contain the same material. The classifier is a linear SVM trained on real images with annotations.

**Strengths:**

- This paper provides a new perspective into understanding the features generated by popular foundation models.
- The writing is very clear and easy to follow.

**Weaknesses:**

- It is not clear to me that the proposed probing method is sufficient in proving that a model understands 3D relations. The biggest assumption made by the paper is that the result of the linear SVM is equal to the actual 3D relation in the image. This is simply not true. First, the linear SVM cannot get 100% on classifying the relations (this number is not mentioned anywhere in the paper). Second, even if the linear SVM gets 100% on the training dataset, it is not guaranteed to work on out of distribution images. Using the result of such a linear probe as a quantitative measure on how well the model understands 3D structure is very unreliable.
- The paper seems to claim that stable diffusion can be used as a way to extract features from images and is better than alternatives like OpenCLIP and DINO. However, I’m not sure whether it is a good idea of adding 8 different noises to the image and average the features. It can be very slow since diffusion is a sequential process and multiple diffusion passes are needed, while OpenCLIP and DINO just need a single forward pass on the original image.
- The comparison between other foundation models and stable diffusion in Table 4 is not entirely fair, since stable diffusion uses grid search to find the best layer for the task, but other models use the last layer features. Although in Table 3 the authors mention that the difference between last layer and best layer is about 4% for two of the tasks. I don’t think it is enough to conclude that grid search is not needed for all tasks.
- The question designed to verify the relations are too simple. For example, in Fig 7 and 8, the generated shadow does not quite match the shape of the object, but the question only cares about whether the shadow belongs to the object, not about whether the shape matches.

**Questions:**

- Please provide accuracy of the linear SVM on the train/val set for each question.
- In the qualitative examples, the inpainting mask is very tight around the desired region (e.g. the shadow of an object). What if a larger region is masked (e.g. the entire ground)? Can stable diffusion still generate shadow in the correct location and direction?

---

> ### Author Response · Authors · 2023-11-21
> **Rebuttal by Authors**
>
> Thanks for your valuable comments. We have addressed them below, and hope to hear back from you if you have further questions!
>
> W1 (SVM training and testing): The train/val sets and test set are disjoint, so the SVM is testing generalization. We agree that more datasets can always be added to test out of (dataset) generalization.
>
> W2 (8 different noises for Stable Diffusion): We follow DIFT[1] to add 8 different noises to get the more stable feature from Stable Diffusion. We set the batch size to be 8 so we only need a single forward pass.
>
> W3 (Grid search for other models): We have now updated results to have a full grid search for all the foundation models we evaluate on (Table 3,4,5,6,7,8,9 in the revision). Stable Diffusion still outperforms all of them.
>
> W4 (Designed questions for different properties): We agree that more detailed questions, such as the shape of the shadow matching the silhouette of the object causing it, are interesting and could be examined in the future. For the moment, the paper is establishing a protocol for probing such properties. We do mention some possible future extensions in the Discussion and Future Work section.
>
> Q1 (Linear SVM train/val accuracy): We will add this to the supplementary of the paper.
>
> Q2 (Larger inpainting mask):  In the example of Figure 1, we cannot mask out the entire ground as we need the shadows of the other objects to indicate the direction of the shadow of the target object. In the revision, we have updated the shadow example of Figure 1 where a larger region is masked out for the inpainting task, but Stable Diffusion still manages to generate shadow in the correct direction.
>
> References:
> [1] Luming Tang, Menglin Jia, Qianqian Wang, Cheng Perng Phoo, and Bharath Hariharan. Emergent correspondence from image diffusion. NeurIPS, 2023

---

### Author Response · Authors · 2023-11-21
**Author Rebuttal by Authors**

We thank the reviewers for recognising the clarity, novelty and significance of the paper. The major concerns of the reviewers are (1) we should also evaluate other generative models, and (2) we should complete the grid search for DINO and CLIP features for all properties.

For (1), as requested we have added an evaluation on the recent strong GAN model VQGAN and updated in the revision. Though we reiterate that the objective of the paper is to provide a protocol for evaluating different models, rather than comprehensively evaluating all models ourselves. For (2) we have completed the grid search of DINO and CLIP features for all tasks and updated in the revision. Updates in the revision are highlighted in blue.

---

### Meta-Review · Area_Chair_BAVG · 2023-12-05

**Metareview:**

This paper proposes a novel protocol to study features learned by a Stable Diffusion model. Specifically, they considered 7 different properties of scenes (e.g., depth, occlusion, material, etc.) and extracted features from a pre-trained Stable Diffusion model by asking designed questions. The obtained features are further used to learn linear SVM classifier to demonstrate the semantic knowledge embedded in those features. All reviewers think the paper introduces a novel perspective of Stable Diffusion models and include extensive experiments. Concerns raised by reviewers include: a) generalization to out-of-distribution images, b) probing questions are too simple and high-level, c) task formulation is vague.

**Justification For Why Not Higher Score:**

This paper proposes a novel perspective to examine features learned by Stable Diffusion models. Though it brought fresh insights about the model, there are unresolved concerns raised by the reviewers. First, reviewer ssWi and n8LA are skeptical about the conclusion claimed by the authors. The performance of linear SVMs learned on top of the features show that Stable Diffusion models can provide relevant knowledge for each chosen scene property, however, I do agree with reviewers that it is hard to claim that Stable Diffusion understands 3D scenes based on these experiments. The claim is not precise enough and does not provide immediate answers as to how to utilize this knowledge other than correctly answering the chosen questions. Based on these considerations, I recommend to reject this paper.

**Justification For Why Not Lower Score:**

N/A

---

### Decision · Program_Chairs · 2024-01-16

Reject